# Itchy E3 Ubiquitin Ligase-Mediated Ubiquitination of Ferritin Light Chain Contributes to Endothelial Ferroptosis in Atherosclerosis

**DOI:** 10.3390/ijms252413524

**Published:** 2024-12-17

**Authors:** Yi Zeng, Shuai Fu, Yuwei Xia, Guoliang Meng, Xiaole Xu

**Affiliations:** Department of Pharmacology, Pharmacy College, Nantong University, Nantong 226001, China; zengyi1980@ntu.edu.cn (Y.Z.); fushuai@stmail.ntu.edu.cn (S.F.); 2023110077@stmail.ntu.edu.cn (Y.X.); mengguoliang@ntu.edu.cn (G.M.)

**Keywords:** ITCH, ferritin light chain, endothelial cells, ferroptosis, atherosclerosis

## Abstract

This research seeks to investigate the function and fundamental mechanisms of Itchy E3 ubiquitin ligase (ITCH), a HECT (homologous to E6AP carboxyl terminus)-type E3 ubiquitin ligase, in endothelial ferroptosis, particularly in the context of atherosclerosis, which has been underexplored. The levels of ITCH protein in the aortas of mice with atherosclerosis were analyzed. Constructs for ITCH RNA interference were generated and introduced into human aortic endothelial cells (HAECs). The findings indicated that ITCH protein expression was elevated in atherosclerotic mice and HAECs exposed to oxidized low-density lipoprotein (ox-LDL). ITCH downregulation significantly mitigated ox-LDL-induced endothelial injury and dysfunction. Reducing ITCH expression inhibited ox-LDL-induced endothelial ferroptosis. This study also revealed that ITCH mediates ox-LDL-induced ubiquitin-dependent degradation of ferritin light chain (FTL) in HAECs. The protective impact of ITCH knockdown against ox-LDL-induced ferroptosis and endothelial injury was reversed by FTL siRNA. Additionally, in vivo experiments showed that inhibiting ITCH reduced atherosclerosis progression and reversed ferroptosis in the aorta, with an associated increase in FTL protein expression in the aortas of mice. This study demonstrates that ITCH interacts with and regulates the stability of the FTL protein via the ubiquitin–proteasome system, contributing to ox-LDL-induced ferroptosis and endothelial cell dysfunction. Targeting components of the ITCH-FTL pathway holds potential as a therapeutic strategy against atherosclerosis.

## 1. Introduction

Atherosclerosis (AS) represents a persistent inflammatory condition affecting blood vessels, primarily distinguished by the manifestation of atherosclerotic plaques and consequential vascular endothelial injury. Atherosclerosis is a multifaceted process that encompasses aberrant lipid metabolism, oxidative stress, inflammation, and additional concurrent mechanisms. Recently, cell death mechanisms in AS have been clarified [1]. A newly recognized type of cell death, ferroptosis, has emerged as significant. Ferroptosis manifests through a two-stage process. The first stage entails cellular iron overload, catalyzing the significant generation of reactive oxygen species through the Fenton reaction. Subsequently, an imbalance occurs within the intracellular antioxidant system, including components such as the SLC7A11-GPX4-GSH axis. This imbalance leads to an excess of phospholipid hydroperoxides, disrupting typical cellular structure and function [1,2]. Ferroptosis fosters AS by expediting dysfunction in vascular cells through lipid peroxidation and by triggering an inflammatory response. However, ongoing research on ferroptosis in AS is in its early stages. Further comprehensive studies are anticipated to uncover and elucidate the currently limited molecular mechanisms of ferroptosis, thereby establishing a stronger scientific foundation for potential clinical applications in preventing and treating AS.

In the past few years, there has been growing recognition of the involvement of E3 ubiquitin ligases in the progression of diverse diseases, including cardiovascular conditions [3]. ITCH is an E3 ubiquitin ligase originally identified at the agouti locus, which controls the coat color of mice. It functions by attaching a substrate and then transferring it to a ubiquitin-carrying E2 ubiquitin-conjugating enzyme. ITCH is also recognized for its involvement in the underlying mechanisms that contribute to cardiovascular and vascular-related health issues. Research has indicated that TXNIP, a protein linked to cardiovascular problems, is controlled by ITCH to undergo polyubiquitination [4]. The activation of ITCH-driven p73 ubiquitination and degradation by Tax1 binding protein 1 leads to BNIP3-triggered apoptosis in cardiac muscle cells, contributing to the onset of heart failure in murine models [5]. Notably, the absence of ITCH causes a transformation of immune cells from a pro-inflammatory to an anti-inflammatory state, effectively safeguarding mice against complications arising from diet-induced obesity [6]. Additionally, the genetic deficiency of ITCH has been observed to decrease atherosclerotic lesion size in Apoe^−/−^ mice. ITCH plays a role in regulating lipid metabolism, which in turn influences the advancement of atherosclerosis by controlling the ubiquitination of SIRT6 and SREBP2 in the liver [7]. Recently, Huang et al. observed elevated levels of ITCH in clinical samples from 10 atherosclerosis patients. Their findings revealed that ITCH facilitated LDL uptake and lipid accumulation in macrophages through the ubiquitination and subsequent degradation of LKB1 [8]. However, ITCH’s role in atherosclerosis has been insufficiently explored, particularly concerning vascular cell mechanisms.

Lately, two consecutive articles have documented the influence of ITCH on the ferritin light chain (FTL) protein within vascular cells. Borkowska et al. revealed that the degradation of ferritin induced by Ang II, resulting in an elevation of highly reactive iron levels, is orchestrated via the JNK/ITCH axis in endothelial cell lines [9]. Similarly, He et al. demonstrated that LPS stimulation prompts the breakdown of FTL via the ubiquitin–proteasome proteolytic pathway, and this degradation process is mediated by ITCH in murine macrophage cells [10]. Ferritin, an intracellular iron storage protein, comprises two subunits: ferritin light chain and ferritin heavy chain. Its primary role involves intracellular iron storage and utilization, guarding cells against harm resulting from iron metabolism imbalances. Notably, ferritin serves as an indicator of cellular ferroptosis. Within this context, FTL assumes responsibility for maintaining the stability of the ferritin complex [11,12,13]. Hence, we hypothesize that ITCH might influence vascular cell ferroptosis in atherosclerosis.

Given that dysfunction of endothelial cells is pivotal in the early stages of lesion development, the objective of this study is to elucidate the biological significance of ITCH in endothelial ferroptosis during atherosclerosis. This investigation encompasses analyses conducted both in vitro and in vivo and seeks to uncover the underlying mechanisms involved.

## 2. Results

### 2.1. ITCH Exhibited Increased Protein Expression in LDLr^−/−^ Mice Subjected to an HFD and Endothelial Cells Treated with ox-LDL

There was a notable increase in atherosclerotic lesion formation in LDLr^−/−^ mice subjected to an HFD, as evidenced by enlarged lesion areas in aortic root sections (Figure 1A). Ox-LDL, which is a form of LDL altered by oxidation, represents a pivotal risk factor in the development of atherosclerosis. Our findings indicate that LDLr^−/−^ mice fed an HFD exhibited a notably elevated serum level of ox-LDL in comparison to the control group (Figure 1B). After a 12-week high-fat diet regimen, the levels of ITCH protein in the aortas of LDLr^−/−^ mice exhibited a significant increase when compared to those on a regular diet, as shown in Figure 1C. In this context, the co-localization of ITCH and CD31, a common endothelial cell marker, appeared to be elevated in AS model mice (Figure 1D,E). Endothelial cell impairment is widely considered the initial stage in the pathological advancement of atherosclerosis. Ox-LDL plays a pivotal role in initiating injury to endothelial cells. Hence, we conducted an evaluation to determine whether ox-LDL could stimulate the expression of ITCH in cultured human aortic endothelial cells (HAECs). Figure 1F shows that administration of ox-LDL at concentrations spanning 25 to 100 μg/mL resulted in a substantial augmentation of ITCH protein abundance. The zenith of ITCH protein expression was observed at the 24-hour mark subsequent to treatment with 100 μg/mL of ox-LDL (Figure 1G).

### 2.2. ITCH Knockdown Inhibits ox-LDL-Induced Endothelial Injury and Dysfunction

To explore the role of ITCH in activated endothelial cells, we inhibited ITCH expression through the transfection of ITCH-specific siRNA prior to stimulation with ox-LDL. After transfecting HAECs with ITCH siRNA, the protein expression of ITCH was effectively diminished in comparison to the control siRNA group in ox-LDL-induced HAECs (Figure 2A). The reduction in ITCH expression through knockdown significantly counteracted the reduction in cell viability and the elevation in LDH release that were evident in ox-LDL-induced HAECs (Figure 2B,C). In addition, downregulation of ITCH expression alleviated the ox-LDL-induced decrease in NO levels and mitigated the adhesion of monocytes to endothelial cells in the presence of ox-LDL stimulation (Figure 2D,E). Moreover, our data demonstrate that siRNA-mediated suppression of ITCH significantly hindered the ox-LDL-induced upregulation of mRNA expressions for TNF-α, IL-6, and IL-1β in endothelial cells (Figure 2F–H). This inhibitory effect was further confirmed by ELISA data, which revealed a noteworthy reduction in the secretion of these pro-inflammatory cytokines induced by ox-LDL upon ITCH knockdown (Figure 2I–K).

### 2.3. ITCH Knockdown Inhibits ox-LDL-Induced Endothelial Ferroptosis

We next determined whether ITCH contributes to ox-LDL-induced endothelial ferroptosis by measuring some ferroptotic markers. As depicted in Figure 3A, the concentration of ferrous iron, an active form of intracellular iron known to facilitate lipid peroxidation and ferroptosis, showed a substantial increase in ox-LDL-induced HAECs. This elevation was effectively counteracted by the downregulation of ITCH. Given that lipid peroxidation is a primary hallmark of ferroptosis, we investigated whether ITCH played a regulatory role in this process. We observed that the suppressive impact of ox-LDL on cellular GSH concentration was partially mitigated upon ITCH knockdown (Figure 3B). Moreover, downregulation of ITCH led to a reduction in the MDA content and generation of lipid ROS and LPO (indicators of lipid peroxidation) within ox-LDL-induced HAECs (Figure 3C–F).

### 2.4. Downregulation of ITCH Expression Grants Protection Against ox-LDL-Induced Endothelial Ferroptosis Through FTL Modulation

As depicted in Figure 4A, the use of siRNA to knock down ITCH led to an increase in FTL protein expression in HAECs treated with ox-LDL. To investigate whether downregulation of ITCH expression confers protection against ox-LDL-induced endothelial ferroptosis through FTL modulation, we used FTL-specific siRNA to knock down FTL expression in cells for 24 h. Western blotting showed that transfection with FTL siRNA successfully reduced FTL protein expression (Figure 4B). FTL siRNA transfection counteracted the reduction in ferrous iron accumulation resulting from ITCH knockdown in cells treated with ox-LDL (Figure 4C). Additionally, the increase in GSH levels and the decrease in MDA, lipid ROS, and LPO content due to ITCH knockdown were notably restored upon transfecting endothelial cells treated with ox-LDL with FTL siRNA (Figure 4D–G). Furthermore, western blot analysis revealed that ITCH knockdown elevated the protein expressions of ferroptosis-related molecules, namely FTH, GPX4, and SLC7A11, in endothelial cells exposed to ox-LDL. This effect was reversed by silencing FTL using siRNA (Figure 4H–J).

### 2.5. Downregulation of ITCH Expression Grants Protection Against ox-LDL-Induced Endothelial Injury and Dysfunction Through FTL Modulation

As depicted in Figure 5A,B, transfecting FTL siRNA nullified the mitigation of endothelial injury, as evidenced by the restoration of cell viability and reduction in LDH release in ox-LDL-treated cells with ITCH knockdown. Furthermore, FTL siRNA transfection significantly reversed the increased NO levels and reduced monocyte adhesion to endothelial cells observed following ITCH knockdown in cells treated with ox-LDL (Figure 5C,D). Additionally, the reduced mRNA levels and release of TNF-α, IL-6, and IL-1β due to ITCH knockdown were notably reversed by FTL siRNA transfection in endothelial cells exposed to ox-LDL (Figure 5E–J).

### 2.6. ITCH Mediates FTL Degradation in Endothelial Cells

Figure 6A data show that after exposure to ox-LDL for 24 h, there was no change in FTL mRNA expression. However, data from Figure 6B show that ox-LDL treatment significantly reduced FTL protein expression at 24 h. This reduction was notably mitigated by MG132, a proteasome inhibitor, but not by treatment with 3-methyladenine (3-MA), an autophagy inhibitor, or chloroquine, a lysosome inhibitor (Figure 6B). These findings suggest that the degradation of FTL in response to ox-LDL is likely mediated through the ubiquitin–proteasome pathway. Given ITCH’s role as a ubiquitin E3 ligase, we investigated its potential involvement in modulating FTL protein degradation in endothelial cells. To explore this, we evaluated the impact of ITCH on FTL protein stability using cycloheximide, a protein synthesis inhibitor. Our findings revealed that ITCH knockdown extended the half-life of native FTL in HAECs, indicating a role for ITCH in diminishing FTL protein stability (Figure 6C). Subsequently, we explored the potential ubiquitination of FTL by ITCH and its consequent impact on FTL stability. Silencing ITCH through siRNA transfection notably reduced the polyubiquitination of FTL in ox-LDL-treated HAECs (Figure 6D). Endogenous ITCH and FTL in HAECs were reciprocally immunoprecipitated under both conditions with and without ox-LDL (Figure 6E,F). This reciprocal interaction provides additional validation for ITCH’s role in reducing FTL.

### 2.7. Knockdown of ITCH Alleviates Atherosclerosis Progression

We administered ITCH-shRNA lentivirus via tail vein injection to elucidate its spe-cific impact on the size and stability of plaques. As anticipated, ITCH protein levels markedly diminished upon knockdown (Figure 7A). Oil Red O staining demonstrated that inhibition of ITCH significantly reduced atherosclerotic plaque formation both in the en-face-prepared aorta and aortic root compared to the model group (Figure 7B–E). Meanwhile, ITCH knockdown significantly decreased the adhesion molecule VCAM-1 level in aortic extracts (Figure 7F). Additionally, ITCH knockdown decreased the mRNA levels of TNF-α, IL-6, and IL-1β in the aortas of AS model animals (Figure 7G–I). In the ITCH knockdown group, serum TG, TC, and LDL-c showed a decreasing trend compared to the model group, but no statistically significant differences were observed (Figure 7J–M), suggesting that the reduction in lesion size due to ITCH knockdown is not significantly linked to lipid regulation.

### 2.8. Knockdown of ITCH Alleviates Aortic Ferroptosis In Vivo

Immunofluorescence analysis of FTL revealed restored expression in CD31-co-stained endothelial cells due to ITCH knockdown (Figure 8A,B). Western blot results further confirmed increased FTL protein expression in the aortas of ITCH knockdown mice (Figure 8C), although FTL mRNA levels remained unchanged across all groups (Figure 8D). Subsequently, we assessed key ferroptosis markers, including iron content, GSH, MDA, and LPO levels, in murine aortic tissue. The findings demonstrated an abnormal elevation of ferrous iron content, MDA, and LPO levels in aortic tissue from the AS model, coupled with a decrease in GSH levels (Figure 8E–H). ITCH downregulation within the model group effectively reversed the levels of these pivotal ferroptosis markers (Figure 8E–H). Additionally, the inhibition of ITCH resulted in heightened protein expressions of key ferroptosis-related molecules, FTH, GPX4, and SLC7A11, within the aortic tissue of mice (Figure 8I–K). This indicates that ITCH contributes to aortic ferroptosis during atherosclerosis progression.

## 3. Discussion

ITCH, a HECT-type E3 ubiquitin ligase, is increasingly recognized as a key regulator in various biological processes for its capacity to ubiquitinate a wide array of target substrates [14,15]. ITCH’s involvement in cardiovascular system pathophysiology has also been reported. In this study, our primary focus was on investigating the role and underlying mechanism of ITCH in endothelial ferroptosis within the context of atherosclerosis. Notably, our data, for the first time, demonstrate that ITCH contributes to ox-LDL-induced ferroptosis in vascular endothelial cells by orchestrating the ubiquitination and subsequent degradation of FTL. Furthermore, we found that ITCH knockdown mitigated atherosclerosis progression and hindered ferroptosis in LDLr^−/−^ mice. This effect was accompanied by elevated FTL protein expression in the aorta.

To begin, our data demonstrates a noticeable increase in the presence of ITCH protein within the aortas of LDLr^−/−^ mice that were subjected to a high-fat diet. This specific murine model conventionally serves as a physiologically pertinent framework for investigating atherosclerosis [16]. Furthermore, within controlled in vitro settings, we observed that ox-LDL effectively elicited an augmentation in ITCH protein expression within cultured endothelial cells. Oxidized LDL (ox-LDL), resulting from the oxidative alteration of native LDL, holds paramount recognition as the pivotal catalyst in atherosclerosis. The uptake of ox-LDL into the vascular wall frequently gives rise to endothelial dysfunction, a factor widely acknowledged for its substantial role in contributing to the development of atherosclerosis [17,18]. In comparison to the control group, exposure to ox-LDL led to reduced cell viability and diminished levels of NO while also enhancing the adhesion of monocytes to endothelial cells and provoking an inflammatory response. These outcomes collectively demonstrate the successful emulation of dysfunctional endothelial cells characteristic of atherosclerosis. Significantly, the suppression of ITCH effectively mitigated the endothelial damage and dysfunction triggered by ox-LDL. Furthermore, in vivo assays also unveiled that ITCH knockdown alleviates aortic pathology, notably marked by a significant reduction in aortic lesion area, as well as decreased levels of VCAM-1 adhesion molecules and pro-inflammatory factors. In line with our current findings, Huang et al. detected increased ITCH expression in clinical samples obtained from patients with atherosclerosis [8]. Stöhr et al. showed that ApoE^−/−^ITCH^−/−^ mice display a decrease in plaque formation and decreased pro-inflammatory marker levels in the aorta [7]. Together, these results robustly imply that ITCH significantly contributes to the advancement of atherosclerosis.

It should be noted that Stöhr et al. found that ITCH knockout improved serum lipid metabolism, which they linked to reduced atherosclerotic plaque formation [7]. However, in our study, ITCH knockdown did not significantly affect blood lipids. This difference likely arises because Stöhr et al. used ApoE-deficient mice, where ITCH knockout reduced serum TC mainly by impairing SREBP2 breakdown in the liver, leading to increased LDL receptor expression and enhanced LDL reuptake [7]. In contrast, we used LDL receptor-deficient mice, so ITCH knockdown had a weaker effect on LDL receptor expression and a less noticeable impact on blood lipids. This also suggests that ITCH may influence atherosclerosis progression through other mechanisms.

Ferroptosis is closely linked to restricted GSH synthesis, disturbances in iron balance, lipid peroxide accumulation, and fatty acid synthesis. These factors are intimately associated with the progression of atherosclerosis [2,19]. Significantly, ox-LDL can induce damage to vascular endothelial cells, closely associating with ferroptosis. The impairment observed in endothelial cells induced by ox-LDL can be notably alleviated through the application of the ferroptosis inhibitor ferrostatin-1 [20]. By assessing comprehensive ferroptosis-related indices (e.g., ferrous iron, GSH, lipid ROS, LPO, MDA content, GPX4, SLC7A11, and FTH1), this research, for the first time, indicates that the suppression of ITCH expression can effectively hinder ox-LDL-induced endothelial ferroptosis in vitro, as well as aortic ferroptosis in atherosclerotic mice. These results strongly imply that ITCH may wield a significant regulatory role in the process of ferroptosis.

Ferroptosis is intimately linked with disruptions across various facets of iron metabolism, encompassing iron acquisition, storage, utilization, and efflux pathways. Consequently, malfunctioning ferritin significantly undermines cellular antioxidative defenses, culminating in the onset of ferroptosis [2,21]. FTL, one of the two subunits comprising ferritin, forms a complex alongside ferritin heavy chains (FTH) to shape the ferritin structure. The FTH subunit exhibits potent ferroxidase activity, driving the oxidation of ferrous iron. FTL contributes essential acidic residues for iron nucleation on the ferritin molecule [11,21]. Mutations in FTL can hinder the dodecahedron architecture of ferritin, disrupting its iron transport function. FTL’s involvement in the Fenton chemical reaction is also notable, underscoring its pivotal role in fundamental cellular processes such as DNA replication, repair mechanisms, and gene modulation [13]. Ferroptosis is closely associated with disturbances in iron metabolism, including iron uptake, storage, utilization, and efflux. Consequently, impaired ferritin markedly diminishes cellular antioxidant capacity, thereby triggering ferroptosis [22]. Moreover, ferroptosis induced by FTL deficiency is implicated in various pathophysiological processes [11,23]. Previous studies have shown that as an E3 ubiquitin ligase, ITCH is responsible for the LPS-induced ubiquitin–proteasome-dependent degradation of FTL in murine macrophage cells [10]. The current study, for the first time, demonstrates that ox-LDL-induced ubiquitin-dependent degradation of FTL is mediated by ITCH in endothelial cells. Meanwhile, it was found that the alleviating effect on ox-LDL-induced ferroptosis and endothelial damage observed after ITCH knockdown was reversed by FTL siRNA. Additionally, the in vivo experiments in this study demonstrated that inhibiting ITCH reduced the progression of atherosclerosis and reversed ferroptosis in the aorta, accompanied by increased FTL protein expression in the aortas of mice. The findings suggest that the beneficial effects of ITCH suppression on atherosclerosis are at least partially due to inhibiting the ITCH-FTL axis, with ITCH-mediated ubiquitination of FTL contributing to endothelial ferroptosis in atherosclerosis.

## 4. Materials and Methods

An expanded Materials and Methods section is available in the Appendix A. This study was conducted according to National Institutes of Health guidelines for the Care and Use of Laboratory Animals and approved by the Animal Care and Use Committee of Nantong University (protocol code S20210309-003, 9 March 2021). Briefly, eight-week-old male LDLr^−/−^ mice were randomly assigned to either the control group or the AS model group. The control group mice were provided with a standard chow diet. In the AS model cohort, the mice were administered a high-fat diet enriched with 0.21% cholesterol and 21% fat (D12079B, Open Source Diets, Research Diets, Inc., Changzhou, China). After 12 weeks, mice underwent an overnight fast, followed by blood collection from the orbital cavity. After that, the mice were euthanized by CO_2_ inhalation, and the aortic tissues were removed from all groups for further measurement.

In ITCH knockdown studies, LDLr^−/−^ mice were randomly allocated into three groups: the control group + lentivirus-scramble-shRNA (control+sh NC), the AS model + lentivirus-scramble-shRNA (AS+sh NC), and the AS model + lentivirus-ITCH-shRNA (AS+sh ITCH) group. Lentivirus-scramble-shRNA and lentivirus-ITCH-shRNA (5′-AATCCAGACCACCTGAAATAC-3′) were purchased from GenePharma (Shanghai, China). Mice simultaneously received injections with sh NC or sh ITCH lentiviral particles (2 × 10^7^ TU/mouse) via the tail vein once every four weeks. After 12 weeks, the mice were sacrificed as mentioned above.

Serum ox-LDL levels were quantified with ELISA kits following the manufacturer’s instructions. The levels of inflammatory cytokines TNF-α, IL-6, and IL-1β were assessed using ELISA kits, according to the manufacturer’s instructions.

Plaque severity was assessed using serial sections from the aortic root, following the method established in our previous study [24]. Serum levels of total cholesterol (TC), triglycerides (TG), low-density lipoprotein cholesterol (LDL-c), and high-density lipoprotein cholesterol (HDL-c) were measured using a biochemical assay kit, as described in our earlier publications [24].

Cryosections of the thoracic aorta were employed for immunofluorescence staining using appropriate antibodies. Fluorescence intensity was observed using a fluorescence microscope and quantified with Image-Pro Plus analysis software version 6.0.

Human aortic endothelial cells (HAECs) were cultured in endothelial cell medium as described in our previously established method [25]. Cells within passages three to eight were utilized for these experiments.

To knock down ITCH, HAECs were transfected with 100 nM of ITCH siRNA or control siRNA (GenePharma RNAi, Shanghai, China) using Lipofectamine 3000, following the manufacturer’s instructions, for a duration of 24 h. The forward primer sequence for ITCH siRNA is 5′-GCUGUUGUUUGCCAUAGAATT-3′, and the reverse primer sequence is 5′-UUCUAUGGCAAACAACAGCTT-3′. To detect the role of ITCH on ox-LDL-induced endothelial injury and ferroptosis, HAECs were transfected with either 100 nM control siRNA (con siRNA) or 100 nM ITCH siRNA for 24 h and subsequently treated with 100 µg/mL ox-LDL for another 24 h. To investigate the role of FTL in the impact of ITCH on ox-LDL-induced endothelial injury and ferroptosis, cells were first transfected with ITCH siRNA, then subsequently infected with 100 nM FTL siRNA for 24 h, and finally exposed to ox-LDL for an additional 24 h. The forward primer sequence for FTL is 5′-CCUGGAGACUCACUUCCUATT-3′, and the reverse primer sequence is 5′-UAGGAAGUGAGUCUCCAGGAA-3′.

Cell viability was determined with an MTT assay following the protocol outlined in an earlier study [26]. Cell injury was determined by assessing LDH release via a colorimetric LDH cytotoxicity detection kit. NO production was assessed using the Griess reaction with an NO detection kit as previously described [27].

Cell adhesion experiments were conducted following previously established methods [28]. Before the end of treatment, cell treatment was discontinued, followed by the addition of calcein-labeled THP-1 monocytes to facilitate binding for one hour. Subsequently, fluorescence intensity was quantified using a microplate reader (Synergy H1, BioTek Instruments Inc., Winooski, VT, USA) at an excitation wavelength of 485 nm and an emission wavelength of 520 nm.

Ferrous iron contents were measured using an iron assay and were performed according to the manufacturers’ instructions as previously described [29].

GSH levels were assessed using a GSH assay kit according to the manufacturer’s guidelines as described previously [30]. The interaction between 5,5′-dithiobis(2-nitrobenzoic acid) and GSH results in the formation of a yellow chromogen, with its absorbance being recorded at 410 nm.

Lipid peroxidation was assessed using the lipid reactive oxygen species (ROS) fluorescent probe C11-BODIPY581/591 (Invitrogen™, Carlsbad, CA, USA) [16]. The data are expressed as a percentage relative to the control. Lipid peroxidase (LPO) was evaluated by determining the intracellular concentration using a commercial kit, adhering to the provided protocol. To measure MDA, a thiobarbituric acid assay kit was used.

Quantitative real-time PCR (qRT-PCR) analysis was conducted following methods established in our previous studies [24,25]. The sequences of the primers utilized were shown in Appendix A.

Western blot and immunoprecipitation analyses were performed following our previously established methods [25]. The antibody information can be found in Appendix A.

### Statistical Analysis

The data are expressed as mean ± standard deviation. Statistical analysis employed GraphPad Prism 8 software. The Shapiro–Wilk test was used to assess the normality of the data. After confirming normal distribution, an unpaired Student’s *t*-test was used to compare two groups and one-way ANOVA with the Newman–Keuls test for multiple groups, with statistical significance defined as *p* < 0.05.

## 5. Conclusions

In conclusion, this research demonstrated for the first time that ITCH interacts with and regulates the stability of the FTL protein via the ubiquitin–proteasome system, contributing to AS risk factor-induced ferroptosis and endothelial cell dysfunction. Therapeutic targeting of ITCH-FTL pathway components has the potential to be against atherosclerosis.

## Figures and Tables

**Figure 1 ijms-25-13524-f001:**
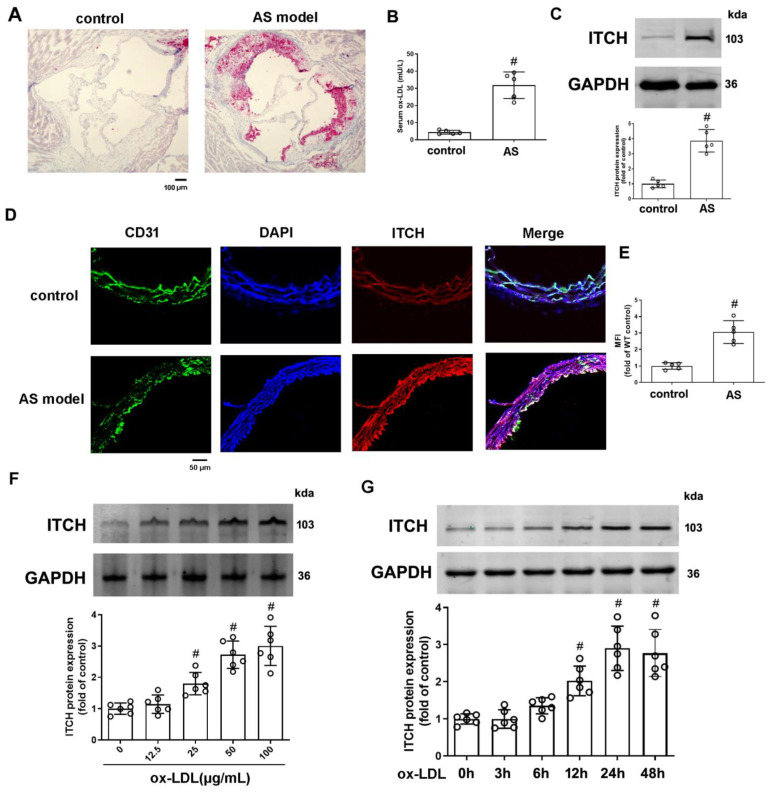
Expression of ITCH in HFD-fed LDLr^−/−^ mice and ox-LDL-treated endothelial cells. (**A**) Representative images of aortic root sections stained with Oil Red O. (**B**) Serum ox-LDL level. (**C**) Protein expression of ITCH in aorta. (**D**) Representative images of thoracic aorta cross sections stained with anti-ITCH antibody and CD31, respectively. (**E**) Quantification of fluorescence intensity of ITCH staining. For (**B**,**C**,**E**), *n* = 5 distinct samples for each group. (**F**) Protein expression of ITCH in HAECs that were incubated with various concentrations of ox-LDL for 24 h. (**G**) Protein expression of ITCH in HAECs that were incubated with 100 μg/mL ox-LDL at various times. For (**F**,**G**), *n* = 6 independent experiments. Results are presented as the mean ± SD. ^#^
*p* < 0.05 vs. control group.

**Figure 2 ijms-25-13524-f002:**
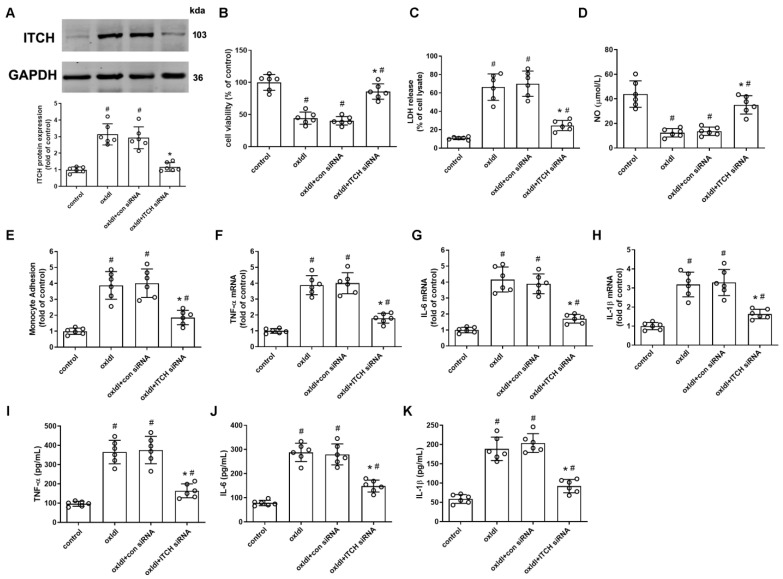
ITCH knockdown inhibits ox-LDL-induced endothelial injury and dysfunction. For (**A**–**K**), HAECs were incubated with control siRNA (con siRNA) or ITCH siRNA for 24 h and then with ox-LDL for an additional 24 h. (**A**) Protein expression of ITCH. (**B**) Cell viability. (**C**) LDH release. (**D**) NO production. (**E**) Monocyte adhesion. (**F**) mRNA expression of TNF-α. (**G**) mRNA expression of IL-6. (**H**) mRNA expression of IL-1β. (**I**) Secretion of TNF-α. (**J**) Secretion of IL-6. (**K**) Secretion of IL-1β. For (**A**–**K**), *n* = 6 independent experiments. Results are presented as the mean ± SD. ^#^
*p* < 0.05 vs. untreated control group, * *p* < 0.05 vs. ox-LDL alone group.

**Figure 3 ijms-25-13524-f003:**
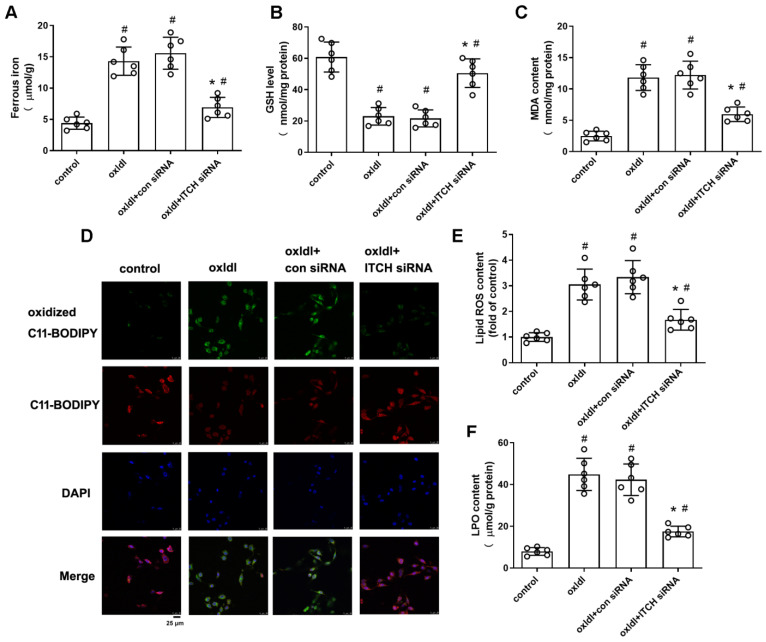
ITCH knockdown inhibits ox-LDL-induced endothelial ferroptosis. For (**A**–**F**), HAECs were incubated with control siRNA (con siRNA) or ITCH siRNA for 24 h and then with ox-LDL for an additional 24 h. (**A**) Ferrous iron contents. (**B**) GSH level. (**C**) MDA content. (**D**,**E**) The lipid ROS levels were determined based on the fluoroprobe C11-BODIPY581/591 by fluorescence microscope (**D**) and fluorescence microplate reader (**E**). (**F**) LPO content. Results are presented as the mean ± S.D. For (**A**–**F**), *n* = 6 independent experiments. ^#^
*p* < 0.05 vs. untreated control group, * *p* < 0.05 vs. ox-LDL alone group.

**Figure 4 ijms-25-13524-f004:**
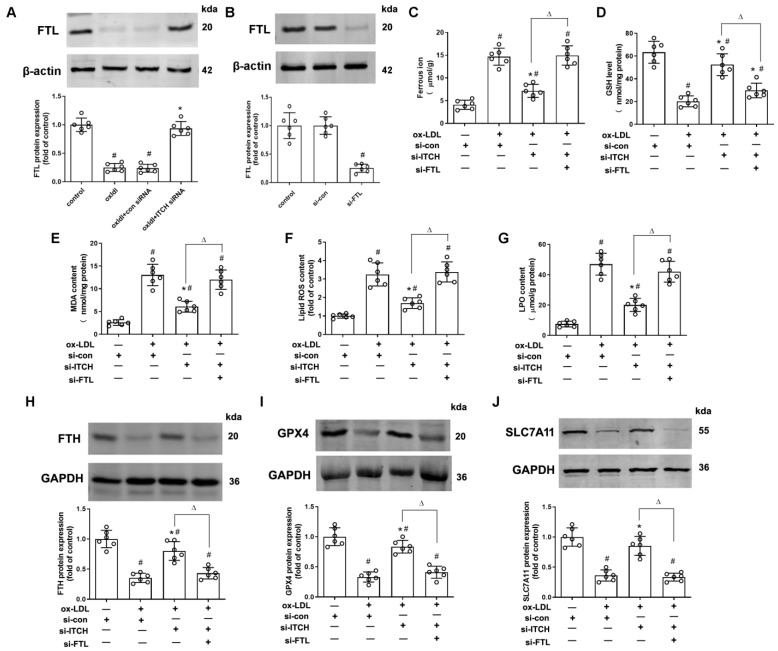
Downregulation of ITCH expression grants protection against ox-LDL-induced endothelial ferroptosis through FTL modulation. (**A**) HAECs were incubated with control siRNA (si-con) or ITCH siRNA (si-ITCH) for 24 h and then with ox-LDL for an additional 24 h. Then, the protein expression of FTL in HAECs was determined. (**B**) HAECs were transfected with si-con or si-FTL for 24 h. Protein expression of FTL was measured. For (**C**–**J**), after transfection with ITCH siRNA, cells were further infected with FTL siRNA for 24 h and then treated with ox-LDL for an additional 24 h. (**C**) Ferrous ion contents. (**D**) GSH level. (**E**) MDA content. (**F**) The lipid ROS levels. (**G**) LPO content. (**H**) Protein expression of FTH. (**I**) Protein expression of GPX4. (**J**) Protein expression of SLC7A11. Results are presented as the mean ± S.D. For (**A**–**J**), *n* = 6 independent experiments. ^#^
*p* < 0.05 vs. si-con group, * *p* < 0.05 vs. ox-LDL+si-con group, ^Δ^ *p* < 0.05.

**Figure 5 ijms-25-13524-f005:**
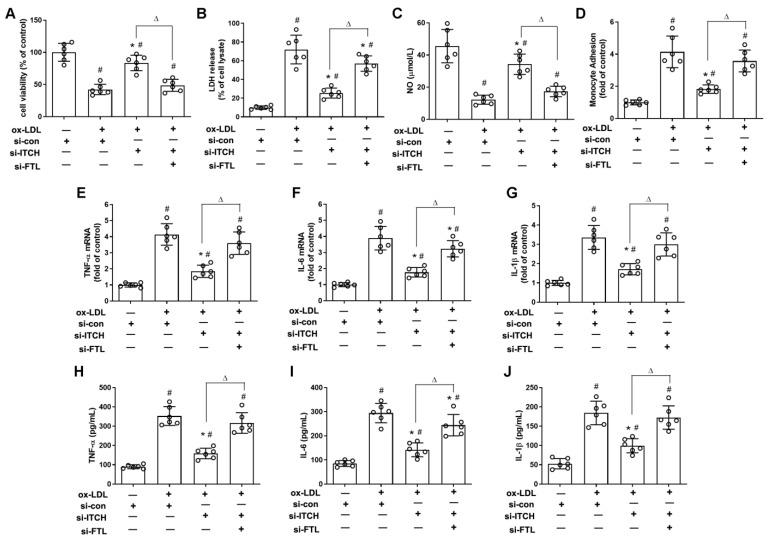
Downregulation of ITCH expression grants protection against ox-LDL-induced endothelial injury and dysfunction through FTL modulation. For (**A**–**J**), after transfection with ITCH siRNA, cells were further infected with FTL siRNA for 24 h and then treated with ox-LDL for an additional 24 h. (**A**) Cell viability. (**B**) LDH release. (**C**) NO production. (**D**) Monocyte adhesion. (**E**) mRNA expression of TNF-α. (**F**) mRNA expression of IL-6. (**G**) mRNA expression of IL-1β. (**H**) Secretion of TNF-α. (**I**) Secretion of IL-6. (**J**) Secretion of IL-1β. Results are presented as the mean ± S.D. For (**A**–**J**), *n* = 6 independent experiments. ^#^ *p* < 0.05 vs. si-con group, * *p* < 0.05 vs. ox-LDL+si-con group, ^Δ^ *p* < 0.05.

**Figure 6 ijms-25-13524-f006:**
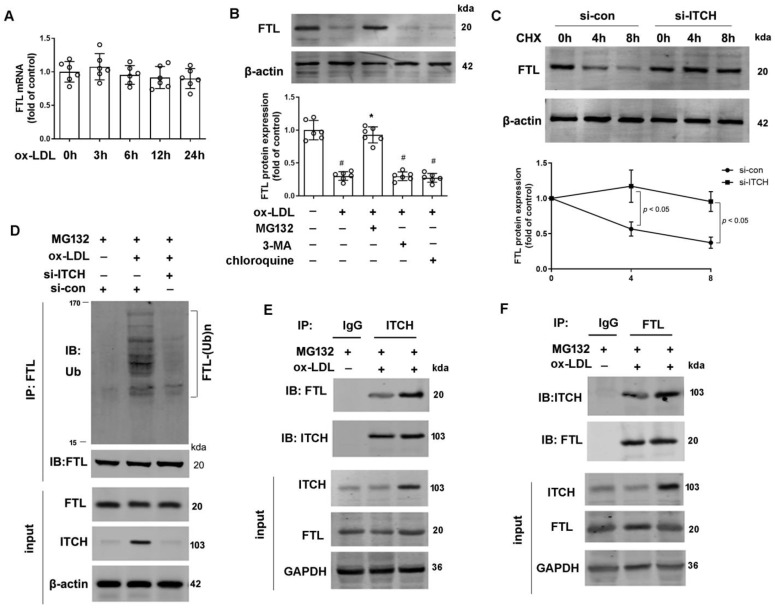
ITCH mediates FTL degradation in endothelial cells. (**A**) mRNA expression of FTL in HAECs that were incubated with 100 μg/mL ox-LDL at various times. (**B**) HAECs pretreated with 3-methyladenine (3-MA, 5 mM), chloroquine (25 μM), or MG132 (20 μM) for 1 h were incubated in the presence and absence of ox-LDL (100 μg/mL) for 24 h. Then, protein expression of FTL was determined. (**C**) HAECs transfected with si-ITCH or si-con in the presence of cycloheximide (CHX, 100 μg/mL) for up to 8 h. Protein expression of FTL was determined. For (**A**–**C**), *n* = 6 independent experiments. (**D**) The ubiquitination of FTL in ITCH siRNA-transfected HAECs under ox-LDL (100 µg/mL) for 24 h in the presence of MG132 (20 µM). (**E**,**F**) HAECs were incubated with 100 μg/mL ox-LDL in the presence of MG132 (20 μM) for 24 h, and cell lysates were then immunoprecipitated to verify endogenous interaction between ITCH and FTL. Results are presented as the mean ± S.D. ^#^ *p* < 0.05 vs. control group, * *p* < 0.05 vs. ox-LDL alone group.

**Figure 7 ijms-25-13524-f007:**
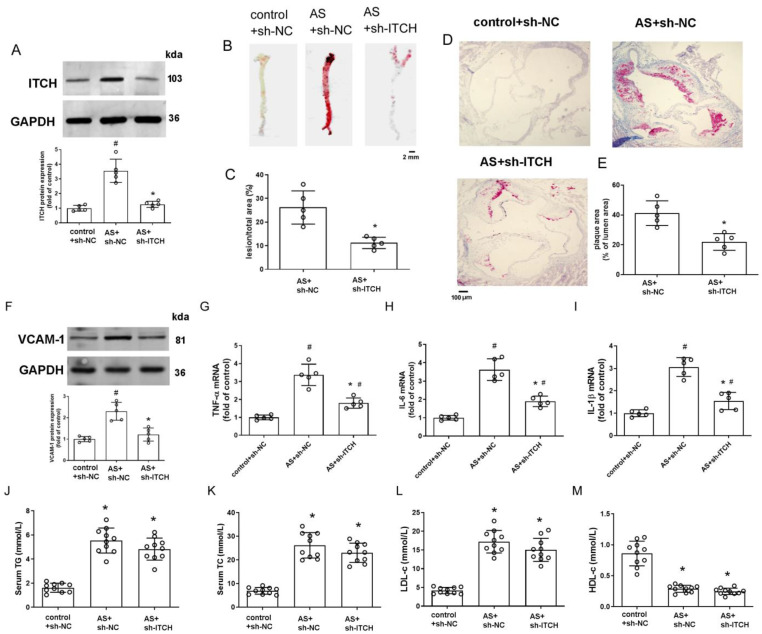
Knockdown of ITCH alleviates atherosclerosis progression and ferroptosis in vivo. (**A**) Protein expression of ITCH in the aorta. (**B**–**E**) Representative images and quantification of en-face preparations of aortas and aortic root sections stained with Oil Red O. (**F**) Protein expression of VCAM-1 in the aorta. (**G**–**I**) mRNA expression of TNF-α (**G**), IL-6 (**H**), and IL-1β (**I**) in the aorta, respectively. For (**A**–**I**), *n* = 5 distinct samples for each group. (**J**) Serum TG level. (**K**) Serum TC level. (**L**) Serum LDL-c level. (**M**) Serum HDL-c level. For (**J**–**M**), *n* = 10 distinct samples for each group. Results are presented as the mean ± SD. ^#^ *p* < 0.05 vs. control+sh-NC group. * *p* < 0.05 vs. AS+sh-NC group.

**Figure 8 ijms-25-13524-f008:**
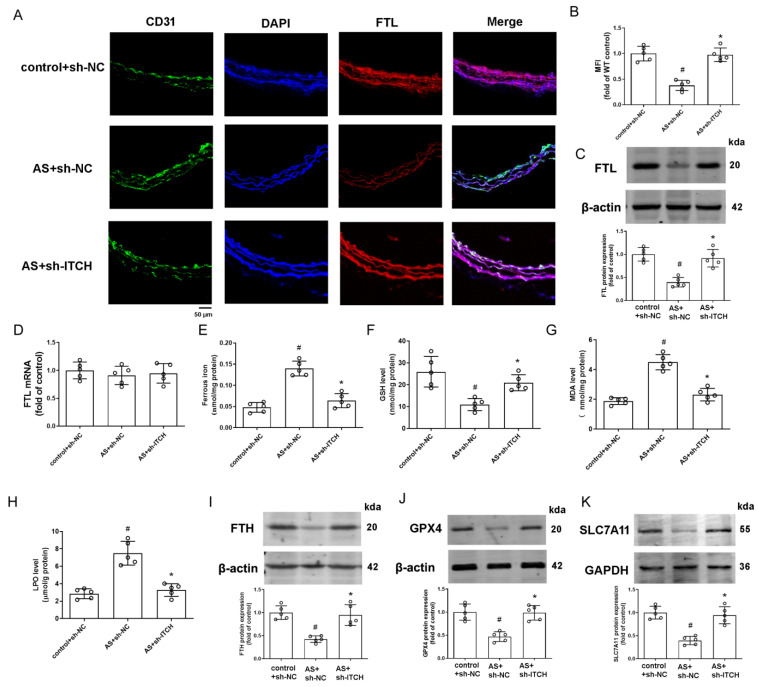
Knockdown of ITCH alleviates aortic ferroptosis and inflammation in vivo. (**A**) Representative images of thoracic aorta cross-sections stained with anti-FTL antibody and CD31, respectively. (**B**) Quantification of fluorescence intensity of FTL staining. (**C**) Protein expression of FTL in the aorta. (**D**) mRNA expression of FTL in the aorta. (**E**) Ferrous iron content in the aorta. (**F**) GSH level in the aorta. (**G**) MDA level in the aorta. (**H**) LPO content in the aorta. (**I**–**K**) Protein expression of FTH (**I**), GPX4 (**J**), and SLC7A11 (**K**) in the aorta, respectively. Results are presented as the mean ± SD. *n* = 5 distinct samples for each group. ^#^ *p* < 0.05 vs. control+sh-NC group. * *p* < 0.05 vs. AS+sh-NC group.

## Data Availability

Data is contained within the article and Appendix A.

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
