# Peer review of "Itchy E3 Ubiquitin Ligase-Mediated Ubiquitination of Ferritin Light Chain Contributes to Endothelial Ferroptosis in Atherosclerosis"

_ijms, 2024, doi:10.3390/ijms252413524_

Round 1
Reviewer 1 Report
Comments and Suggestions for Authors
In the present manuscript, the authors investigated the role if ITCH in endothelial cells exposed to oxidized LDL in vitro as well as in atherosclerosis development in vivo. Inhibition of ITCH reduced endothelial ferroptosis both in vitro and in vivo. This is associated with increased availability of the FTL protein. The study is interesting, and the manuscript is well written. However, there are major concerns that must be addressed by the authors:
1) The method section in the supplement is insufficient as there are several missing informations. It is very important to provide as many details as possible so that other researchers can reproduce your findings:
a) Animal Experiments: The authors need to include a statement that the protocol was approved by the Institutional Animal Care and Use Committee and that all animal experiments are conducted in accordance with established animal welfare guidelines and regulations.
b) Provide catalog # for LDLR-/- mice and for standard chow diet.
c) Provide name, company, catalog #, dilution of the antibodies used for immunofluorescence staining of aorta.
d) Provide catalog # of HAECs used for cell culture. e) Provide catalog # for NO detection kit. f) Provide name, company, catalog #, dilution of the antibodies used for Western blot analyses. g) Provide catalog # of lysis buffer used for Western blot.2) The description of statistical analysis is insufficient:
a) The authors need to include a statement regarding number n of experiments for cell culture (independent replicates?) and animal studies (e.g. 5 experiments from 5 different animals).
b) The authors didn't perform normality test for such small sample sizes. If the data don't pass normality test, then they must be analyzed by non-parametric tests, i.e. Mann-Whitney test, etc.
3) Figure legends: Are the numbers n of experiments same in all groups? If so then the authors need provide more precise information, e.g. n=x per group.
4) The authors used bar graphs for presentation of results in the figures. Bar graphs are applicable for frequencies in categorical variables only but not for presenting continuous/quantitative measures. Therefore, the results need to be presented in box plots or dot plots instead of bar graphs. (Weissgerber TL et al. PLoS Biol. 2015 Apr 22;13(4):e1002128).
Author Response
Comments 1: The method section in the supplement is insufficient as there are several missing informations. It is very important to provide as many details as possible so that other researchers can reproduce your findings:
a) Animal Experiments: The authors need to include a statement that the protocol was approved by the Institutional Animal Care and Use Committee and that all animal experiments are conducted in accordance with established animal welfare guidelines and regulations.
b) Provide catalog # for LDLR-/- mice and for standard chow diet.
c) Provide name, company, catalog #, dilution of the antibodies used for immunofluorescence staining of aorta.
d) Provide catalog # of HAECs used for cell culture. e) Provide catalog # for NO detection kit. f) Provide name, company, catalog #, dilution of the antibodies used for Western blot analyses. g) Provide catalog # of lysis buffer used for Western blot.
Response 1: Thank you for your comments. We have added the statement about animal experiments in the Methods and Materials section of the revised manuscript. Catalogs for LDLR-/- mice, standard chow diet, antibodies used for immunofluorescence staining of aorta, HAECs, NO detection kit, antibodies and lysis buffer used for Western blot have been added in the Supplemental Materials and Methods section.
Comments 2: The description of statistical analysis is insufficient:
a) The authors need to include a statement regarding number n of experiments for cell culture (independent replicates?) and animal studies (e.g. 5 experiments from 5 different animals).
b) The authors didn't perform normality test for such small sample sizes. If the data don't pass normality test, then they must be analyzed by non-parametric tests, i.e. Mann-Whitney test, etc.
Response 2: Thank you for your comments. The number (n) of experiments for cell culture and animal studies has been added to the figure legends of each panel. When analyzing small sample data with fewer than 50 samples, we used the Shapiro-Wilk test for normality. The Shapiro-Wilk test indicated that all data in the article follow a normal distribution. In the revised manuscript, we have presented this result in the Statistical analysis section.
Comments 3: Figure legends: Are the numbers n of experiments same in all groups? If so then the authors need provide more precise information, e.g. n=x per group.
Response 3: Thank you for your comment. We have provided more precise information in the figure legends, specifying n = x per group. However, since another reviewer felt that the figure legends were too wordy, we have combined the descriptions for figures with the same sample size.
Comments 4: The authors used bar graphs for presentation of results in the figures. Bar graphs are applicable for frequencies in categorical variables only but not for presenting continuous/quantitative measures. Therefore, the results need to be presented in box plots or dot plots instead of bar graphs. (Weissgerber TL et al. PLoS Biol. 2015 Apr 22;13(4):e1002128).
Response 4: Thank you for your comment. We have replaced the bar graphs with dot plots as requested. However, Figure 6C is not suitable for a dot plot, so we prefer to keep it in the XY plot format.
The revisions to the manuscript have been highlighted using the "Track Changes" function in MS Word.
Reviewer 2 Report
Comments and Suggestions for Authors
The authors investigated the fundamental function and mechanisms of ITCH, endothelial ferroptosis, especially in the context of atherosclerosis.
The introduction is well structured with suggestive information for the topic studied.
In the results, the figure legends are presented in too much detail. I recommend using keywords in the figure legends and explanations in the text.
Why did the experimental model last 12 weeks?
The conclusions are clear and based on the results obtained. However, I recommend that Fig.9 not be integrated into the conclusions.
Author Response
Comments 1: In the results, the figure legends are presented in too much detail. I recommend using keywords in the figure legends and explanations in the text.
Response 1: Thank you for your comments. Based on your suggestion, we have appropriately shortened the figure legends for Figures 1 to 6, removing content that has already been explained in the methods section.
Comments 2: Why did the experimental model last 12 weeks?
Response 2: LDLr-/- mice typically develop significant atherosclerotic plaques in the aorta and coronary arteries after 12 weeks of high-fat feeding
Comments 3: The conclusions are clear and based on the results obtained. However, I recommend that Fig.9 not be integrated into the conclusions.
Response 3: Thank you for your comments. We have removed Figure 9 in the revised manuscript.
The revisions to the manuscript have been highlighted using the "Track Changes" function in MS Word.
Round 2
Reviewer 1 Report
Comments and Suggestions for Authors
Thank you for providing detailed information.